# Advancing urban management: Integrating GIS, LLMs, and media narratives into environmental and socio-economic analyses for enhanced urban crime analysis

Yuxuan Cai[1,☯,*], Anzhi Chen[1,☯,*], Ziheng Tang[2,‡], Yuhan Wang[2,‡], Yang Song[2]

1 Social Science Division, University of Chicago, Chicago, Illinois, United States of America,
2 Department of Landscape Architecture & Urban Planning, Texas A&M University, College Station, Texas, United States of America

☯ These authors (Yuxuan Cai and Anzhi Chen) contributed equally to this work.
‡ These authors (Ziheng Tang and Yuhan Wang) contributed equally to this work.
* yuxuanc@uchicago.edu (YC); chenanzhi@uchicago.edu (AC)

## Abstract

This study examines how crime is reported in the media, how the public perceives crime, and how urban environmental and economic factors interact at both global and local spatial scales to shape criminal activity. The research analyzed 3,507 reports from the top 100 media outlets from 2023 to 2024 and utilized a large language model to extract key information. The test results showed that the inter-coder reliability between human labeling and LLM labeling datasets reached 0.92 and examining the discrepancies between media narratives and actual crime data. By applying traditional econometric and Geographic Information Systems (GIS) geospatial regression techniques, the study revealed the impacts of environmental and economic factors on crime distribution, as well as the spatial relationships between urban structural features (such as road network density and vacant buildings) and crime rates in the Chicago area. Additionally, through Geographically Weighted Regression (GWR), the study further investigated the local differences between crime rates and socio-economic and environmental factors. The research found that selective news reporting might distort public understanding and affect policy responses. The findings provide insights for urban safety strategies and crime prevention policies, emphasizing the need to enhance the accuracy of media reports and public education, and offer valuable guidance for policymakers and urban planners on how to improve urban safety through environmental and economic planning.

**Data availability statement:** The environmental and economic data underlying the results presented in this study are available from the Chicago Data Portal (https://data.cityofchicago.org/), the United States Environmental Protection Agency (https://www.epa.gov/smart-growth/smart-location-mapping#SLD), and the News database via the Perigon API (https://www.goperigon.com/).

**Funding:** The author(s) received no specific funding for this work.

**Competing interests:** The authors have declared that no competing interests exist.

# 1. Introduction

## 1.1. Media perception of crime

In urban settings, media narratives significantly shape public perceptions of crime, influencing collective consciousness, policymaking, and community dynamics [1,2]. Beyond mere reporting, media representations actively construct societal realities by selectively emphasizing certain crimes and framing them in ways that affect how communities interpret and respond to criminal activities [5–7]. News consumers often regard media reports as highly trustworthy, which can distort public perceptions by sensationalizing crime and creating a gap between perceived and actual crime rates [3,4,7]. This discrepancy contributes to heightened public fear of crime, particularly amplified in the digital era through constant exposure to crime-focused content on social media and online news platforms [8–10]. Additionally, media portrayals frequently reflect and amplify societal biases, disproportionately highlighting crimes in specific communities or involving individuals from certain racial or ethnic backgrounds, thereby perpetuating stereotypes and social inequalities [11–15].

## 1.2. Urban Built Environment with Crime

A wide range of studies have explored the effects of various factors such as urban built environment, economic environment, human activities, and their perception of crime incidents. Kim and Wo focused on studying the impact of the physical environment on walkability-related crimes [16]. Moreover, He et al., combining the framework of environmental criminology with an integration of urban micro-environmental characteristics and macroeconomic features, proposed a comprehensive assessment framework that more effectively explains the occurrence of crime [17]. Building on this, examining how geographic space influences the variability of crime distribution provides a fresh perspective for assessing crime characteristics on a spatial scale [18].

## 1.3. Research gaps

In Chicago, previous crime studies have often focused on features or dimensions, such as socioeconomic or urban environmental factors with overall crime events. Hou et al. found that high temperatures significantly increase urban crime risk in Chicago [19], while Ogletree et al. observed a significant positive correlation between urban green spaces and crime rates in Chicago [20]. Schusler et al. further explored the impact of tree canopy coverage on crime in Chicago [21]. From the perspective of urban economic activity, Walter et al. discovered a significant negative correlation between building permit activity, property changes, and violent crime [22]. While the impact of urban features and dimensions on crime has been widely studied, few have undertaken a comprehensive investigation incorporating data from subjective media reporting, objective crime events, and socioeconomic and environmental factors— particularly in the context of the Chicago area. Given that the occurrence of crime is influenced by multiple factors and the complex interactions between environment and crime, as well as potential spatial heterogeneity, conducting a crime study in

the Chicago area to explore the discrepancies between subjective reporting and objective events, as well as the global and spatial correlations of the built environment and socio-economic conditions with urban crime becomes especially important.

By analyzing news articles from major Chicago media outlets, the study delves into the portrayal and contextualization of various crimes, such as Assault, Criminal Damage, and Theft. This aspect is crucial as it uncovers the narrative frameworks and recurring themes in media reporting, providing insights into how societal perceptions of crime are shaped. Such an understanding is vital for policymakers and social scientists alike, as it reflects the interplay between media narratives and public opinion. This qualitative analysis, combined with quantitative data, offers a comprehensive view of the crime scenario, bridging the gap between different research methodologies in social science. This research project represents a significant advancement in computational social science, particularly in understanding urban crime dynamics in Chicago.

## 2. Data and methods

### 2.1. Study Area & Research Framework

Chicago, as the third largest city in the United States, has historically been a crucial site for studying the relationship between urban economic environments and crime rates. With its highly diverse ethnic population, complex social structures, economic conditions, and distinct crime patterns, the city offers a unique platform for in-depth exploration of how various socioeconomic factors and urban environmental elements interact with urban crime activities. In recent years, despite making some progress in reducing crime and improving the safety of its inhabitants, certain areas of the city continue to suffer from high crime rates, significantly exceeding those in other U.S. cities and the average in Illinois.

To study crime, we used a dataset of crime in the city of Chicago from 2017 to 2023 downloaded from the Chicago Data Portal. Crime data provides the type of crime, date, and geographic coordinates for each incident. A total of 234,918 crimes are collected.

The Study utilizes data at the census tract level, including summaries of geo-point data within census tracts, as the basis for spatial analysis. This scale, chosen based on previous research, includes detailed population and economic information, enabling a more accurate exploration of the relationship between economic variables and crime. Variables related to the built environment are primarily focused on geo points and polygons, aggregated to the census tract scale to facilitate this investigation. (Fig 1)

The analysis focuses on three distinct aspects. First, we select the three most common crimes for detailed examination based on the actual crime rate: 1. theft; 2. battery; 3. criminal damage. (Fig 2) Second, we explore the subjective narratives of media reports and the disparity between media-favored crime and the most frequent crime in the Chicago area. This involves comparing the state of distribution of subjective reports to the objective reality of actual crimes. Finally, we discuss the relationship between crime rates and both the built environment and socioeconomic factors, exploring how urban form and socio-economic characteristics impact urban crime.

### 2.2. Use of LLM in media reporting

We employ a two-stage approach to analyze crime reporting trends across top media sources. Initially, we utilize the Perigon database to systematically extract news articles published by the top 100 media outlets from the year 2023–2024. This news dataset serves as the corpus for our analysis, ensuring a broad and representative sample of crime reporting from a diverse array of geographical and editorial perspectives. As a proprietary AI-powered news analytics platform that aggregates articles from media outlets and extracts structured event information – including geographic coordinates, event types, and summaries – using natural language processing and proprietary classifiers, it filters out opinion or gossip content, ensuring relevance to crime reporting. The dataset does not include every local press piece as we select the top 100 media sources, making it well-suited for our analysis of media narratives around crime [23]. Meanwhile, another

| Media Perspectives | Environment and Socioeconomic Measurement | | | Crime Measurement |
|---|---|---|---|---|
| *Perigon Data* | *United States Environmental Protection Agency* | *United States Census Bureau* | *Open Street Map* | *Chicago Data Portal* |
| GPT4.0 + GIS Analysis | | GIS Analysis | | GIS Analysis |

| **New Report** | **Environment Factor** **Crime Prevention through Environmental Design (CPTED)** | **Socioeconomic Factor** | | **Crime Data** |
|---|---|---|---|---|
| Top 100 News Media 3507 Reports | Territoriality : Number of POI (Count_of_P) | Total Population (Population) | | 2017-2023 Year 234918 Reports |
| | Surveillance : Total Road Network Density (Total_road) | | | |
| | Activity Support : Prevalence of vacant and abandoned buildings (Building_C) | Housing Units (Housing_un) | | |
| | Image Maintenance: walkability,total water area (Total_wate), total water area (Total_wate) | Number of Households (Households) | | |

| RQ1: | RQ2: | RQ3: |
|---|---|---|
| **Large Language Model (LLMs)** | **Spearman Correlation and Multiple OLS model** | **GWR(Geographically Weighted Regression)** |
| Data Mining and Crime Identification | Regression | Regression |
| The gap between media coverage and reality | Correlations between environmental, socio-economic and three type crime evens. | Spatial correlation and heterogeneity |

Implement

| whether media outlets prioritize certain locales and types of crime over others. | Urban Planning Suggestions |
|---|---|

**Fig 1. Research Design and Framework.**

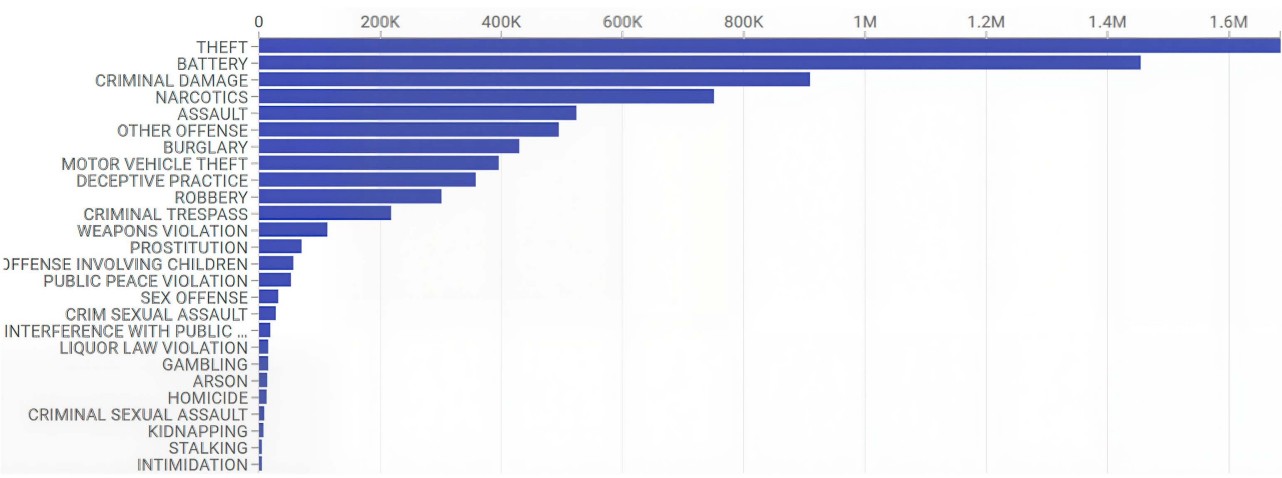

**Fig 2. Summary of the number of Crime incidents.**

advantage of this dataset is that it provides the location of the crime if the location has been covered by the original article. The selection criteria for these media sources are based on their reach and credibility, as indexed by Perigon, to ensure that our analysis encompasses a wide spectrum of reporting styles and audience engagements. In total, the Perigon database returned 3,507 articles.

Following the data collection phase, we leverage the advanced natural language processing capabilities of GPT-3.5-Turbo, a variant of the generative pre-trained transformer models developed by OpenAI, to categorize reported crimes into designated categories. Text classification has always been the central focus of LLM models and shows exceptional capacities in text analysis [24]. In this study, we further utilize GPT-3.5-Turbo's capability as a transfer learning model developed for natural language problems, as Brown et al. described it as a few-shot learning [25,26]. To initiate the few-shot learning, we selected 10 articles for each crime category – such as theft, battery, criminal damage, narcotics, assault, robbery, weapons violation, sex offense, and others – as initial examples. The categorization of reported crimes into designated categories follows the Chicago Open Data crime classification schema, which is based on the Illinois Uniform Crime Reporting system [27]. This approach ensures consistency between our media-based dataset and official crime data. Within these categories, we focused on three major crime typologies – theft, criminal damage, and battery – because they were the most frequently recorded offenses in the Chicago Open Data crime dataset (Fig 2). This alignment is critical for enabling meaningful comparative and spatial analysis. Then we expanded the model's learning by introducing an additional set of 10 articles per category. The iterative process was designed to refine the model's predictive accuracy and adaptability by exposing it to a broader range of linguistic expressions and crime scenarios.

With 20 examples per category, the model was then tasked with classifying the remaining corpus. Each article was processed through GPT-3.5 Turbo, which utilized the patterns learned from the labeled examples to categorize new instances. The effectiveness of this classification was evaluated by comparing the model-generated labels against a validation set previously categorized by human experts. between the AI and human annotations and to validate the model's performance. We assessed inter-coder reliability between human annotators and GPT-3.5-Turbo using Cohen's kappa ($\kappa$), a statistic that adjusts for agreement expected by chance [28]. Cohen's $\kappa$ ranges from –1 (complete disagreement) to +1 (perfect agreement), with values above 0.80 regarded as 'almost perfect,' whereas $\kappa = 0.92$ indicates exceptionally high agreement between the model and human coders [29]. This high level of inter-coder reliability validates the efficacy of the few-shot learning approach in our study, ensuring that the automated classifications are consistent with expert human judgments.

The utilization of GPT-3.5-Turbo in this context not only enhances the accuracy and efficiency of data analysis but also provides a scalable framework for further comparing subjective media data with the crime data obtained from section 1.

## 2.3. Environmental, socioeconomic factors and crime

To export the relationship between environmental and socio-economic variables with the three distinct crime typologies, this study went through basic correlation detection and classical metrology model. Then it employs Moran's I statistics to discern potential spatial correlations and harnesses Geographically Weighted Regression (GWR) to probe the spatial heterogeneity manifested by different variables. The discourse extends to evaluating how the architectural and economic milieu exerts an influence on the empirical occurrences of crime. Since the number of theft crimes (Theft, number of criminal damage crimes (Criminal Damage), and number of battery crimes (Battery) are the highest in the dataset, this study will only focus on the following three.

### 2.3.1. Census Level data: Environmental and Socioeconomic Measurement.
Drawing from Crime Prevention through Environmental Design (CPTED) theory, this paper acquires six environmental variables across four distinct environmental dimensions (Fig 3): Territoriality, Surveillance, Activity Support, and Image Maintenance. The data is sourced from the environmental variables provided by the Smart Location Database, and Chicago Data Portal. Point of Interest (POI) data retrieved from OpenStreetMap and includes a broad range of urban features such as restaurants,

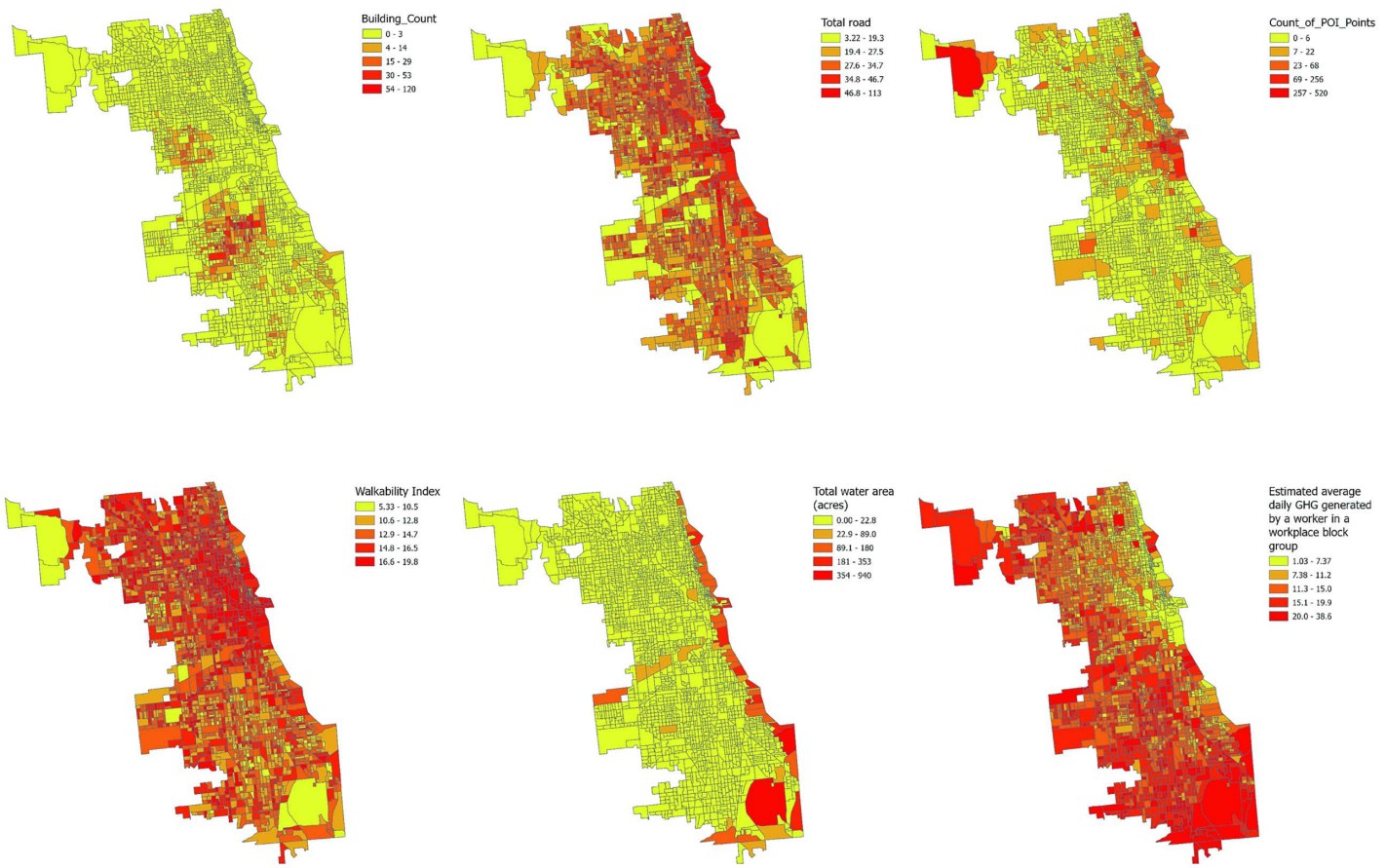

**Fig 3. Spatial Distribution of Environment Variables.**

retail stores, service facilities, and landmarks. Some census tracts – particularly in the downtown business district – have POI counts exceeding 400, which reflects actual high commercial density rather than data errors. We retained all POIs to comprehensively capture territoriality and activity support in line with CPTED principles, even though certain POIs may have varying relevance to crime dynamics.

Territoriality is interpreted here through the number of POI, which indirectly indicates how spaces are utilized and frequented. A higher number of POIs can suggest more vibrant and actively managed spaces, potentially enhancing the sense of territoriality as these areas are likely to be better maintained and monitored by stakeholders. Some studies demonstrate that street connectivity is an important physical environmental feature for understanding criminal opportunities and guardianship [30,31].

Then surveillance is assessed by the total road network density (Total Road), which can affect the natural surveillance of an area by increasing sightlines and potential informal supervision by residents and passersby. Activity Support is represented by the prevalence of vacant and abandoned buildings (Building Count), as these can affect the perceived and actual usage of spaces, potentially impacting the crime rates due to decreased natural surveillance and territorial definition. Image Maintenance encompasses walkability, indicated by the National Walkability Index score, and the total water area (Total Water, in the rest of this article we will use Total Water to replace total water area), both contributing to the aesthetic and functional qualities of an environment. Also, walkability serves as a proxy for environmental maintenance quality, reflecting infrastructure investment and community care [32].

Additionally, the study considers the estimated average daily greenhouse gas emissions generated by a workplace block group (Estimated GHG), as a proxy for activity levels and economic vitality, which may correlate with crime occurrences due to differential levels of place usage and associated informal surveillance.

Total water area was included as an indicator within the Image Maintenance dimension of CPTED, capturing aesthetic and structural qualities of the urban environment. In Chicago, this measure primarily reflects Lake Michigan, the riverfront, and associated open spaces that shape the city's urban fabric. These areas contain major recreational and mobility infrastructures such as public beaches (e.g., Oak Street Beach, North Avenue Beach) and continuous lakefront trails, including bike lanes connecting South Loop and Hyde Park. While water features may appear disproportionately large in spatial maps, their inclusion represents actual geographic characteristics relevant to urban design. In non-coastal cities, analogous measures would include large reservoirs, river corridors, or urban green-blue infrastructure, serving the same conceptual purpose of capturing spatial openness and environmental quality. This represents an exploratory part, as while existing blue space literature extensively examines the health benefits and socio-economic impacts of urban water bodies (rivers, lakes, coastlines), there is limited empirical investigation into how blue space accessibility and environmental quality influence crime rates and terrestrial criminal activity patterns [33,34].

Meanwhile, GHG were incorporated as a proxy for urban economic and activity intensity, extensive prior research linking emissions with economic activities [35,36]. Also, for sectoral and daily activities, Activity-based models show that shifts in daily routines – such as working from home versus commuting to an office – result in measurable changes in GHG emissions [37]. While natural systems also emit GHGs, the portion from human activities (industry, transport, energy) is significant and has been increasing, making anthropogenic GHG emissions a reliable indicator of human-driven activity [38]. Even though this is the case, we do acknowledge that they do not uniformly capture all activity types. For example, high-emission industrial zones differ from commercial areas in their relationship to pedestrian flows and crime risk. Future work could refine this measure by integrating direct mobility metrics, such as transit ridership or anonymized mobile device data.

Socio-economic characteristics have been extensively examined, with numerous studies demonstrating significant correlations between variables such as income, unemployment, housing conditions, and crime rates [39,40]. Thus this paper selects three primary variables as economic indicators from the American Community Survey Data to delve into this relationship further: Total Population (Population), Housing Units, and Number of Households (Households). (Fig 4)

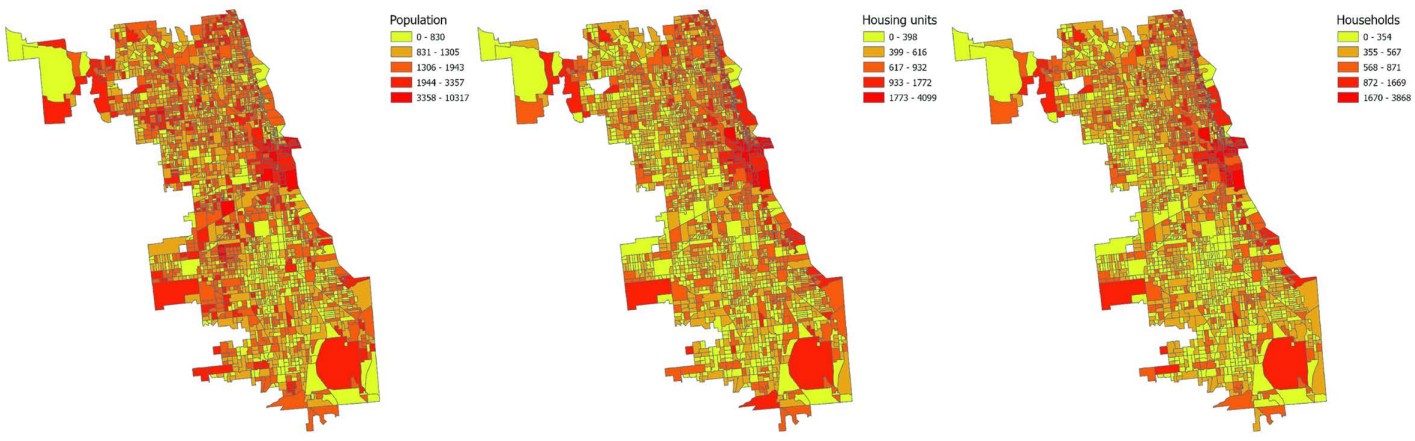

**Fig 4. Spatial Distribution of Socio-economic Variables.**

## 2.4. Spatial autocorrelation and geographically weighted regression

The models discussed in the previous sections are fundamentally global models, and thus, it has been demonstrated that they are incapable of effectively capturing the local or spatial patterns between environmental and socio-economic factors and crime events. Since spatial autocorrelation can influence linear models' performance, we used global Moran's I (Moran, 1950) to examine whether the blocks have the issue of spatial autocorrelation [41].

The models discussed in the previous sections are fundamentally global models, and thus, it has been demonstrated that they are incapable of effectively capturing the local or spatial patterns between environmental and socio-economic factors and crime events. In assessing the spatial distribution characteristics of theft, criminal damage, and battery cases, Moran's I statistics provided evidence of spatial autocorrelation at varying degrees for these types of crimes (Fig 5). Specifically, theft cases exhibited a moderate level of spatial autocorrelation (Moran's I = 0.5044, $p < 0.001$) (Table 1). In the case of criminal damage and battery, Moran's I value rose to 0.7581 and 0.7145, respectively, suggesting a more pronounced spatial clustering. This pattern indicates that the distribution of these crimes is far from uniform, but rather densely occurs in areas with specific urban structures and socio-economic characteristics. This further emphasizes the non-randomness of crime occurrences and their association with specific spatial factors, they also lay the groundwork for subsequent GWR analysis.

By calculating Local Moran's I for each area, we can categorize each observation point into one of four quadrants: High-high (HH): Indicating that an area with a high crime rate is surrounded by other areas with high crime rates, suggesting that certain urban regions may have universally elevated crime rates due to specific factors. This is denoted by "1."Low-low (LL): Where an area with a low crime rate is surrounded by others with low crime rates, suggesting these regions are relatively safer or have effective crime prevention measures in place, denoted by "2."High-low (HL) and

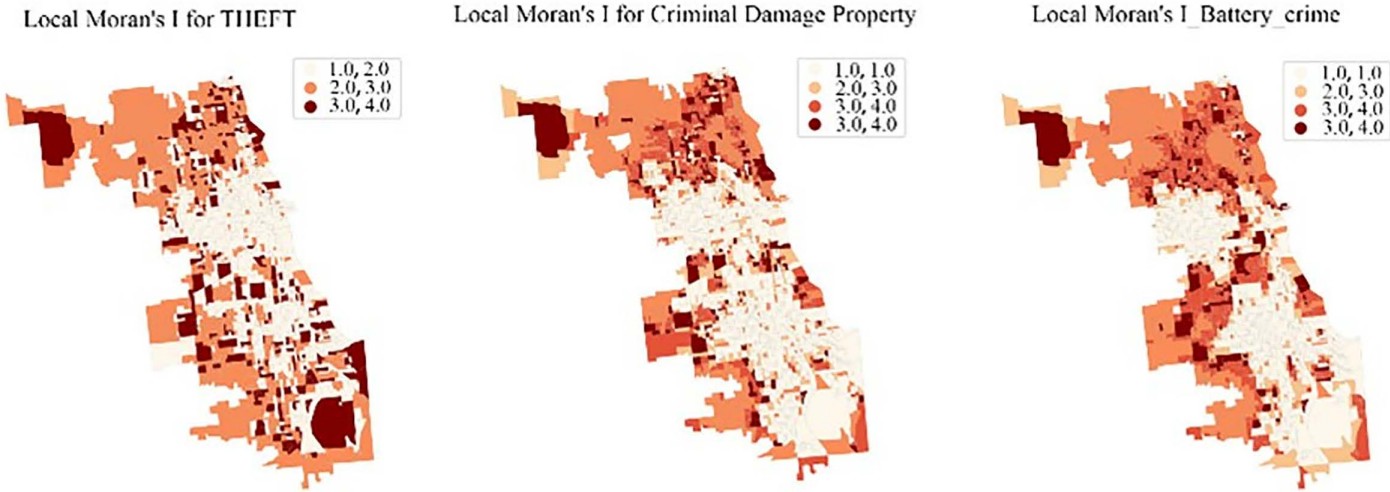

**Fig 5. Local Moran's I.**

**Table 1. Moran's, I Result.**

| Crime | Moran's I | p-value |
|---|---|---|
| Theft | 0.504 | 0.001 |
| Criminal Damage | 0.758 | 0.001 |
| Battery | 0.714 | 0.001 |

Low-high (LH): These patterns typically indicate spatial anomalies in crime rates where an area significantly differs from its neighboring regions, which may be attributed to particular socio-economic variables or urban planning issues. These are represented by "3" and "4" respectively. Each quadrant offers a distinct insight into the spatial dynamics at play, which is crucial for targeted intervention and strategic resource allocation. function, expressed as follows:

$$I_i = \frac{n}{\sum_j w_{ij}} \times \frac{\sum_j w_{ij} (x_i - \bar{x})(x_j - \bar{x})}{\sum_j (x_j - \bar{x})^2},$$

(1)

where $I_i$ is the Local Moran's I value for location $i$, n is the total number of observations. $w_{ij}$ are the spatial weights between locations $i$ and $j$ determined by a spatial contiguity relationship such as Queen contiguity. $x_i$ and $x_j$ is the observed value at location $i$ and $j$. $\bar{x}$ is the mean of the observed values.

The identification of significant spatial autocorrelation in crimes such as theft, criminal damage, and battery reinforces the notion that urban crime is a phenomenon dependent on spatial context. This insight adds a crucial dimension to our understanding of urban crime dynamics, advocating for policies and interventions tailored to different spatial dimensions. Past research has already demonstrated that community-scale surveillance can effectively reduce crime rates. As cities continue to evolve, integrating spatial analysis into urban planning and public safety initiatives remains critical to creating safer and more resilient urban environments.

Leveraging the benefits of spatial clustering, we further employed Geographically Weighted Regression (GWR) to investigate explanatory power within different scale regions. Employing the Variable Inflation Factor (VIF) test to remove variables with high collinearity is a key step to ensure the accuracy of the GWR model; here, variables with a VIF greater than 10 were removed.

In this GWR model, we can see function 2,

$$y_i = \beta_{0(u_i,v_i)} + \beta_{1(u_i,v_i)} x_{i1} + \beta_{2(u_i,v_i)} x_{i2} + \cdots + \beta_{k(u_i,v_i)} x_{ik} + \epsilon_i,$$

(2)

where $y_i$ represents the dependent variable, namely the crime rate in a specific urban area $i$; $x_{ij}$ denotes the independent variables at observation $i$. corresponding to selected factors; the coefficients $\beta_k$ quantify the influence of each independent variable on the dependent variable; and $\epsilon_i$ encapsulates the random error, accounting for the variability not explained by the model. The spatial coordinates $(u_i, v_i)$ of each observation point are utilized to compute distances to other points, which in turn dictate the allocation of the weight function.

One of the quintessential features of the GWR model is its spatial weighting function, which adjusts the influence of each data point on parameter estimation based on the relative distance between observations. This ensures that points closer to the target observation exert a greater influence on the local parameter estimates than those further away. To achieve this, our study opted for the widely used Gaussian function as the weighting function, expressed as follows:

$$w_{ij} = exp\left(-\frac{d_{ij}^2}{2s^2}\right),$$

(3)

where $w_{\{\}}$ represents the weight $d_{ij}$ is the distance between observation points $i$ and $j$ and the bandwidth parameter $s$ controls the rate of decay of weights with increasing distance. Moreover, the "Golden search" optimization algorithm was deployed to determine the optimal bandwidth parameter $s$ critical step for ensuring that the weighting function appropriately reflects the spatial relationships between observation points. By leveraging this approach, the GWR model offers a unique perspective for evaluating and interpreting the crime phenomena within the urban environments of Chicago. This analysis reveals the extent to which different neighborhoods may deviate from city-wide trends, thereby providing

 

a rich spatial context that is essential for targeted policy-making and strategic planning in crime prevention and urban development.

Four core variables were selected: the number of Housing Units (Housing Unit), walkability (walkability), estimated greenhouse gas emissions (Estimated GHG), and the count of points of interest (POI). We found the model to exhibit similar and high explanatory power across different types of crime data. Specifically, for theft cases, the adjusted $R^2$ value was 0.7062. In the analysis of cases of criminal damage, the adjusted $R^2$ value was 0.6956, while for cases of battery, the adjusted $R^2$ value was 0.6786, all indicating high explicability, especially when considering spatial variations. This evidently reveals the heterogeneous spatial patterns of crime events in relation to environmental and economic elements.

## 3. Results

### 3.1. A Comparison between media data and crime data

Our analysis results show the media coverage reveals distinct patterns in crime reporting that diverge significantly from actual crime statistics recorded in the Chicago Open Data Portal. The media demonstrates a clear preference for certain crime types, with robbery, homicide, and assault receiving disproportionate attention relative to their actual frequency.

From Figs 6–7, it's observed that prevalent crimes such as criminal damage and battery do not receive proportional attention in media reports, with newspapers infrequently covering these incidents. This disparity is identical to Professor Lundman's theory that news media frequently have "inordinate attention" to homicide and have always been the most frequently reported type of crime in the Chicago region [42,43]. Even though the literature still lacks concrete evidence on the elements influencing the decision-making process regarding the coverage, and extend of coverage, of specific homicide, we suspect that crimes like robbery, homicide, and assault, possibly due to their more immediate and visceral threat to public safety, might be deemed more newsworthy [44]. The definition of newsworthiness can be traced to Surette's definition, the standard of criteria that news producers use to select which among all known events should be presented to the public as news [45]. This editorial choice reflects a prioritization that could skew public perception, suggesting that the media's portrayal of criminal activity does not always align with actual crime statistics and trends, which is proved in our analysis.

Upon closer examination of the media coverage patterns in relation to the spatial distribution of crime in Chicago, it is evident that newspaper reports predominantly focus on the downtown vicinity, followed up by the O'Hare International

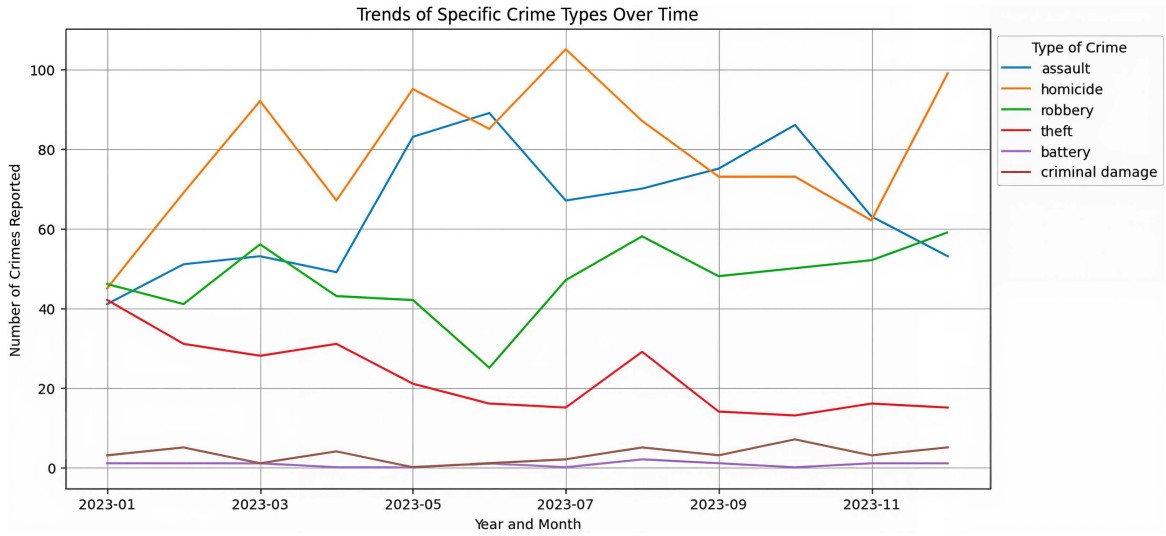

**Fig 6. Most frequently reported crime from major media sources.**

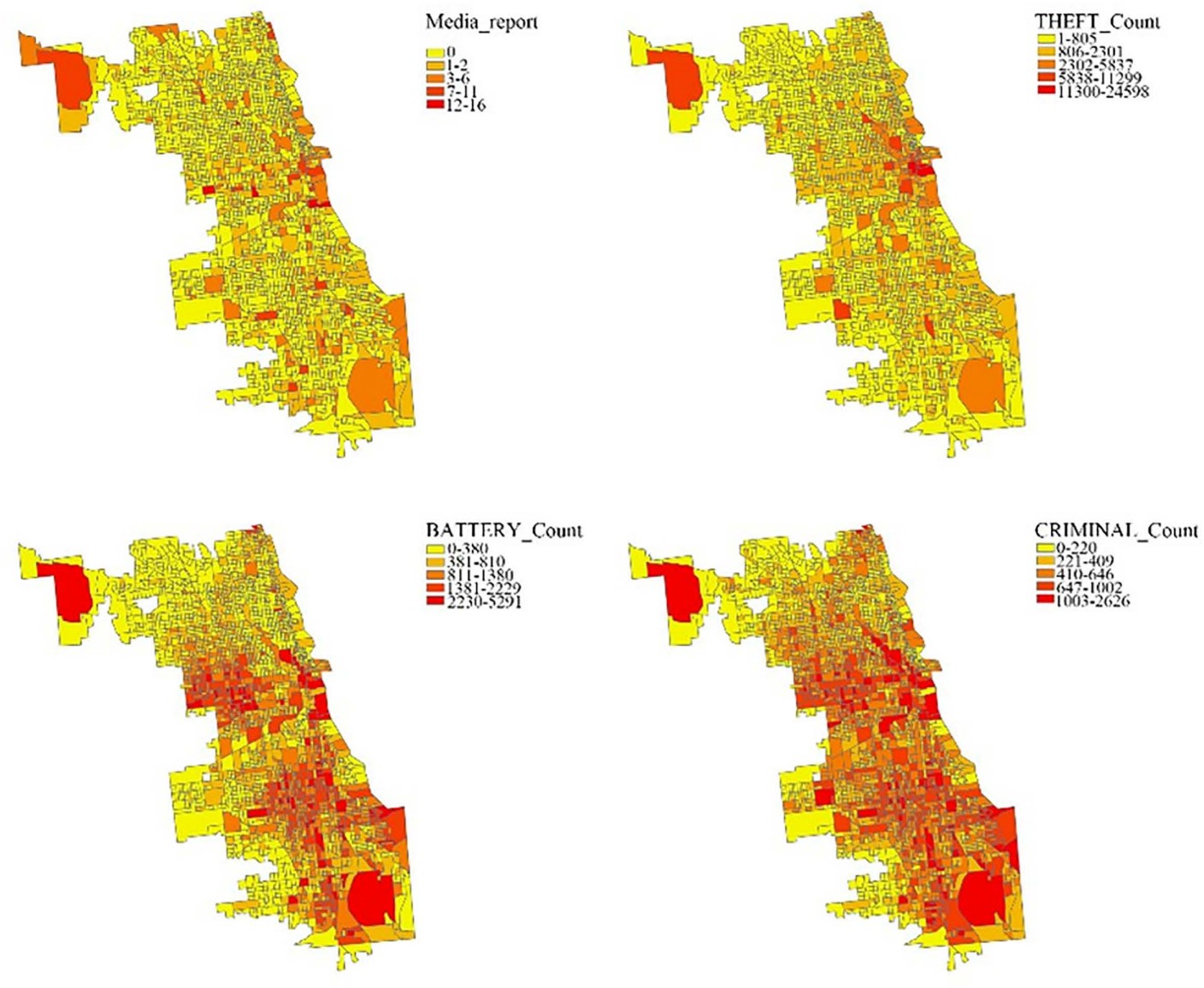

**Fig 7. Spatial distribution of media reports and three crimes.**

Airport area. The latter is considered as predominated by white people in 50.1%, followed up by 23.4% Hispanic people and 16.4% black people, according to Community Data Snapshot O'Hare [46]. Several quantitative research have concluded that reasons leading to media coverage being disproportionately increased compared with actual data include victims being white or higher social status [43,47–49]. Additionally, there is a noticeable emphasis on the South Holland and South Chicago regions, notably the South Shore community. This pattern, though bears a resemblance to the distribution of theft incidents as recorded in the Chicago Data Portal, differs from other dimensions. Therefore, it can be seen as a stigmatization towards the South Shore community produced from denigration of spaces [50]. Since Wacquant first defined the term territorial stigmatization, several articles have explored how the South Side of Chicago has been stigmatized by the policy resulted from racist discrimination and segregation; this part will be detailed discussed in section 3.3.

Meanwhile, a marked disparity exists in the reporting of battery and criminal damage incidents, which are considerably underrepresented in the media. Consequently, the spatial distribution of crimes reported by newspapers diverges significantly from the actual data provided by the Chicago Open Data Portal. This gap highlights a critical area for future research, necessitating a deeper investigation into the mechanisms and editorial decisions that lead to the selective coverage of certain types of crimes, particularly battery and criminal damage, over others.

### 3.2. Correlation analysis for variables

In conducting normality tests for variables including total road network density, walkability, vacant and abandoned buildings, number of POI, Total Water, Total Population, Housing Unit, Households, Estimated GHG, Number of Theft, Number of Criminal Damage crime, and Number of Battery, it is observed from the data presented above that the sample size for each variable exceeds 50 (Table 2). Therefore, the Kolmogorov-Smirnov (K-S) test is employed for this analysis. Specifically, these variables all exhibit significant results ($p < 0.05$). This implies the rejection of the null hypothesis, which posits that the data is normally distributed. Consequently, it can be concluded that none of the variables exhibit characteristics of normality.

Since none of the variables exhibit characteristics of normal, we utilize Spearman correlation analysis to examine the relationships between the three main crime incidents and nine other environmental and economic factors (Table 3).

In the exploration of the nexus between urban attributes and crime incidents, our analysis reveals significant correlations that elucidate the multifaceted impact of urban design and environmental factors on crime rates. Specifically, theft incidents (Theft) exhibit a notable positive correlation with both road network density and walkability index, with correlation coefficients of 0.429 and 0.266, respectively, at a significance level of 0.01. This indicates that urban areas characterized by a denser road network and enhanced walkability may inadvertently provide greater opportunities for theft, underscoring the influence of urban design and accessibility on the occurrence of crime.

Furthermore, the presence of vacant and abandoned buildings is positively correlated with theft rates, with a correlation coefficient of 0.412, further accentuating the role of urban decay in the dynamics of crime. Conversely, an inverse relationship is observed with Total Water area and Estimated GHG with correlation coefficients of −0.041 and −0.524, respectively, suggesting that waterfront environments have minimal correlation with crime, and lower greenhouse gas emissions

**Table 2. All the variables result of normality test (N = 3992).**

| Name | Average | SD | Skewness | Kurtosis | Kolmogorov-Smirnov Test | | Shapiro-WilkTest | |
|---|---|---|---|---|---|---|---|---|
| | | | | | D-value | p | W-value | p |
| Total Road | 27.015 | 8.625 | 0.770 | 4.608 | 0.036 | 0.000** | 0.967 | 0.000** |
| Walkability | 13.639 | 2.920 | −0.704 | 0.378 | 0.073 | 0.000** | 0.966 | 0.000** |
| Building | 1.179 | 5.090 | 9.288 | 132.159 | 0.422 | 0.000** | 0.240 | 0.000** |
| POI | 3.447 | 13.992 | 22.338 | 686.798 | 0.403 | 0.000** | 0.180 | 0.000** |
| Total Water | 4.287 | 37.727 | 16.493 | 329.790 | 0.463 | 0.000** | 0.089 | 0.000** |
| Population | 1308.547 | 632.033 | 2.135 | 14.859 | 0.084 | 0.000** | 0.886 | 0.000** |
| Housing Unit | 548.195 | 294.987 | 3.050 | 20.187 | 0.117 | 0.000** | 0.798 | 0.000** |
| Households | 491.751 | 268.117 | 2.861 | 18.537 | 0.116 | 0.000** | 0.814 | 0.000** |
| Estimated GHG | 14.998 | 5.541 | 0.090 | −0.288 | 0.015 | 0.039* | 0.997 | 0.000** |
| Theft | 417.011 | 958.769 | 12.024 | 235.152 | 0.332 | 0.000** | 0.373 | 0.000** |
| Criminal Damage | 226.756 | 283.343 | 1.580 | 3.833 | 0.227 | 0.000** | 0.797 | 0.000** |
| Battery | 363.214 | 566.273 | 2.384 | 7.501 | 0.261 | 0.000** | 0.691 | 0.000** |

* $p < 0.05$ ** $p < 0.01$ Sample size: 3992

**Table 3. Spearman correlation.**

| | 1 | 2 | 3 | 4 | 5 | 6 | 7 | 8 | 9 | 10 | 11 | 12 |
|---|---|---|---|---|---|---|---|---|---|---|---|---|
| Theft (1) | 1 | | | | | | | | | | | |
| Criminal Damage (2) | 0.947** | 1 | | | | | | | | | | |
| Battery (3) | 0.938** | 0.982** | 1 | | | | | | | | | |
| Total Road (4) | 0.429** | 0.417** | 0.406** | 1 | | | | | | | | |
| Walkability (5) | 0.266** | 0.187** | 0.159** | 0.490** | 1 | | | | | | | |
| Building Count (6) | 0.412** | 0.533** | 0.540** | 0.160** | −0.050** | 1 | | | | | | |
| POI (7) | 0.536** | 0.447** | 0.433** | 0.336** | 0.445** | 0.077** | 1 | | | | | |
| Total Water (8) | −0.041** | −0.060** | −0.068** | −0.162** | −0.113** | −0.062** | −0.007 | 1 | | | | |
| Population (9) | −0.029 | −0.044** | −0.089** | −0.127** | −0.011 | −0.091** | 0.024 | 0.155** | 1 | | | |
| Housing Unit (10) | 0.127** | 0.061** | 0.031* | −0.064** | 0.043** | −0.116** | 0.137** | 0.180** | 0.810** | 1 | | |
| Households (11) | 0.046** | −0.033* | −0.068** | −0.102** | 0.041** | −0.193** | 0.107** | 0.183** | 0.847** | 0.968** | 1 | |
| Estimated GHG (12) | −0.524** | −0.429** | −0.434** | −0.363** | −0.439** | −0.011 | −0.471** | 0.102** | −0.086** | −0.278** | −0.240** | 1 |

\* $p < 0.05$ \*\* $p < 0.01$

may represent key factors in reducing crime rates. Lower emissions often serve as a proxy for reduced economic activity and may also reflect higher environmental consciousness and community quality. These communities are likely more prosperous, with stronger social cohesion and higher levels of surveillance, fostering an environment where crime is more naturally deterred, and community members engage in mutual vigilance. Previous research has shown that communities with environmentally friendly policies have a significant effect on reducing crime rates [51].

Criminal Damage demonstrates a robust positive correlation with building count (correlation coefficient = 0.533), reinforcing the notion that neglected urban areas are more susceptible to higher crime rates. Negative correlations with Total Water and Population (correlation coefficients of −0.060 and −0.044, respectively) imply that broader demographic and environmental characteristics significantly influence the spatial distribution of criminal damage in complex and variable manners.

Battery incidents are significantly positively correlated with total road and building count (correlation coefficients of 0.406 and 0.540, respectively), indicating that the urban fabric and the condition of the built environment are pivotal factors in the prevalence of such crimes. In contrast, significant negative correlations with Total Water and Population suggest a nuanced relationship between crime rates and specific environmental and demographic variables, indicating that more densely populated areas and those with certain environmental features may experience lower rates of battery.

In sum, the density of the road network, the number of vacant and abandoned buildings, and the quantity of Points of Interest (POIs) exhibit stable and significantly positive correlations with all three types of crime studied, while the impact of water-front environments appears to be minimal. Furthermore, lower levels of greenhouse gas emissions may indicate key socioeconomic factors conducive to crime reduction. These findings highlight the critical role of urban planning and environmental management in crime prevention and reveal the complex influences of population density and environmental quality on crime rates, emphasizing the importance of holistic approaches in addressing urban crime.

Utilizing OLS regression, the study meticulously evaluates the influence of urban infrastructure—surveillance (total road network density), activity support (vacant and abandoned buildings), and image maintenance (Walkability, Total Water, Estimated GHG emissions, Population, Housing Units, Households)—on the incidence rates of theft, criminal damage, and battery. The regression outcomes, including coefficients, standard errors, and p-values, elucidate the intensity and significance of these relationships (Table 4).

Specifically, areas with denser road networks showed higher crime rates, likely due to increased anonymity and more escape routes for perpetrators. Similarly, the presence of vacant and abandoned buildings predicted higher crime occurrences, indicating that neglected spaces are potential hotspots for criminal activities. Interestingly, walkability exhibited

**Table 4. Ordinary least squares for three type crime.**

| | Theft | | | Criminal Damage | | | Battery | | |
|---|---|---|---|---|---|---|---|---|---|
| | Coef. | S.E. | p | Coef. | S.E. | p | Coef. | S.E. | p |
| Intercept | −14.319 | 84.650 | 0.866 | 166.740 | 30.240 | *** | 386.810 | 61.340 | *** |
| Total Road | 8.997 | 1.395 | *** | 5.147 | 0.498 | *** | 7.166 | 1.011 | *** |
| Walkability | 5.826 | 4.293 | 0.175 | −0.732 | 1.534 | 0.633 | −8.981 | 3.111 | ** |
| Building Count | 15.375 | 2.058 | *** | 14.750 | 0.735 | *** | 30.717 | 1.492 | *** |
| POI | 42.582 | 0.772 | *** | 3.631 | 0.276 | *** | 6.496 | 0.560 | *** |
| Total Water | 0.520 | 0.273 | 0.057 | 0.287 | 0.097 | ** | 0.407 | 0.198 | * |
| Population | 0.044 | 0.029 | 0.127 | 0.085 | 0.010 | *** | 0.117 | 0.021 | *** |
| Housing Unit | 2.070 | 0.190 | *** | 1.329 | 0.068 | *** | 3.242 | 0.138 | *** |
| Households | −2.003 | 0.221 | *** | −1.609 | 0.079 | *** | −3.804 | 0.160 | *** |
| Estimated GHG | −17.729 | 2.260 | *** | −9.928 | 0.807 | *** | −14.321 | 1.638 | *** |
| R^2 | 0.553 | | | 0.347 | | | 0.328 | | |
| Adjusted R^2 | 0.552 | | | 0.346 | | | 0.326 | | |
| p | *** | | | *** | | | *** | | |

* p<0.05 ** p<0.01 ***p<0.001

a complex relationship with crime; it marginally increased theft incidents but inversely correlated with violent crimes like battery, suggesting that increased pedestrian traffic might deter more severe crimes through enhanced visibility and social interaction.

Additionally, environmental features such as total water areas showed a nuanced positive correlation with crimes like battery and criminal damage. Higher GHG emissions, typically a proxy for industrial or commercial activity, unexpectedly correlated negatively with theft rates, possibly reflecting that economically vibrant areas might deter petty crimes. Lastly, demographic factors such as population density, housing units, and households positively influenced crime rates, affirming that urban congestion provides a conducive environment for criminal activities. These findings underscore the intricate interplay between urban environmental characteristics and crime dynamics, offering insights into urban planning and crime prevention strategies.

The model exhibits varying degrees of explanatory power across the three crime types, with theft presenting the most explicable variance adjusted R² 0.552, notably, factors such as total road network density, the number of buildings, number of Poi count, Housing Units, and Households demonstrate a significant positive association with theft rates, with most variables being significant at the 0.001 level. In contrast, the model's explanatory prowess is comparatively lesser for criminal damage, with Adjusted R² values at 0.346, key predictors include the number of Buildings, Total Water, Population Density, Housing Units, and Households, with the number of buildings exerting a particularly substantial influence.

For battery, the Adjusted R² is 0.326, Significant variables impacting Battery rates include the number of Buildings, Parks, Housing Units, and Households, with the number of buildings exhibiting the strongest positive correlation. The statistical significance of these variables, except for walkability, accentuates their influence on battery occurrences.

The findings underscore the multifaceted nature of urban crime and the pivotal role played by various facets of urban infrastructure in either mitigating or exacerbating crime rates. Specifically, the study calls for a critical examination of road network planning to ensure that increased surveillance does not inadvertently elevate crime risks. Moreover, addressing the blight of vacant and abandoned buildings through re-vitalization initiatives could significantly dampen crime rates.

The mixed impact of image maintenance factors on different crime types further stresses the importance of tailored urban design strategies that not only enhance urban aesthetics and environmental sustainability but also fortify community safety.

### 3.3. Geographically Weighted Regression Reveals Spatial Dynamics

The GWR analysis results reveal the distribution of influence coefficients for four distinct variables on three types of crime within the Chicago context, as well as the spatial variability of these influences at the census tract level. (Fig 8–9 and Fig 10)

**3.3.1. Downtown Chicago.** Result from the downtown Chicago area demonstrate a significant negative correlation between the number of Housing Units, greenhouse gas emissions, and the number of points of interest with the frequency of crime incidents. This suggests that higher residential density and a vibrant array of points of interest may lead to enhanced community surveillance and a heightened sense of public safety. The confluence of dense economic activity and foot traffic could also lead to increased investment in safety measures and police presence, thereby effectively reducing

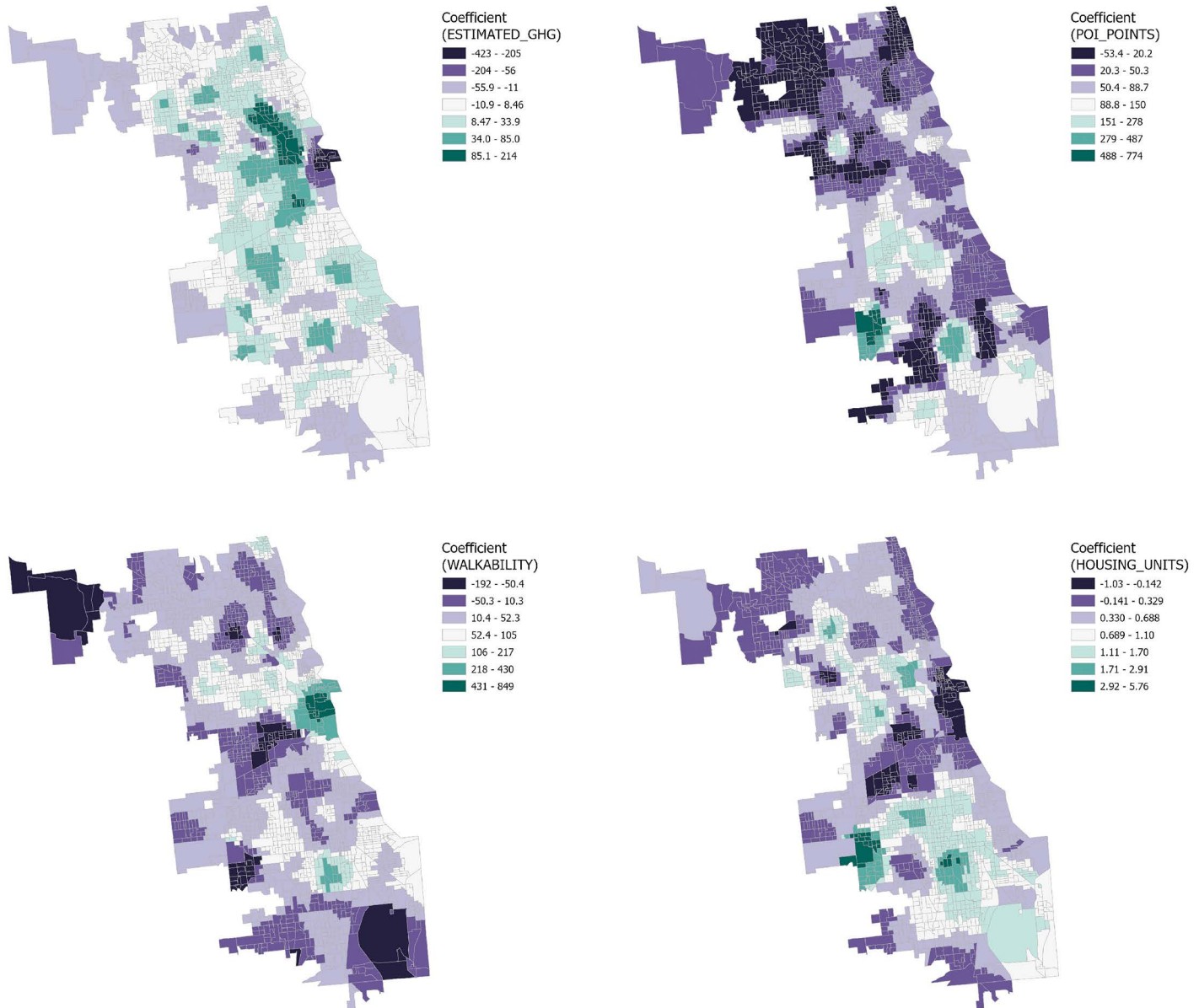

**Fig 8. GWR coefficient distribution of Theft.**

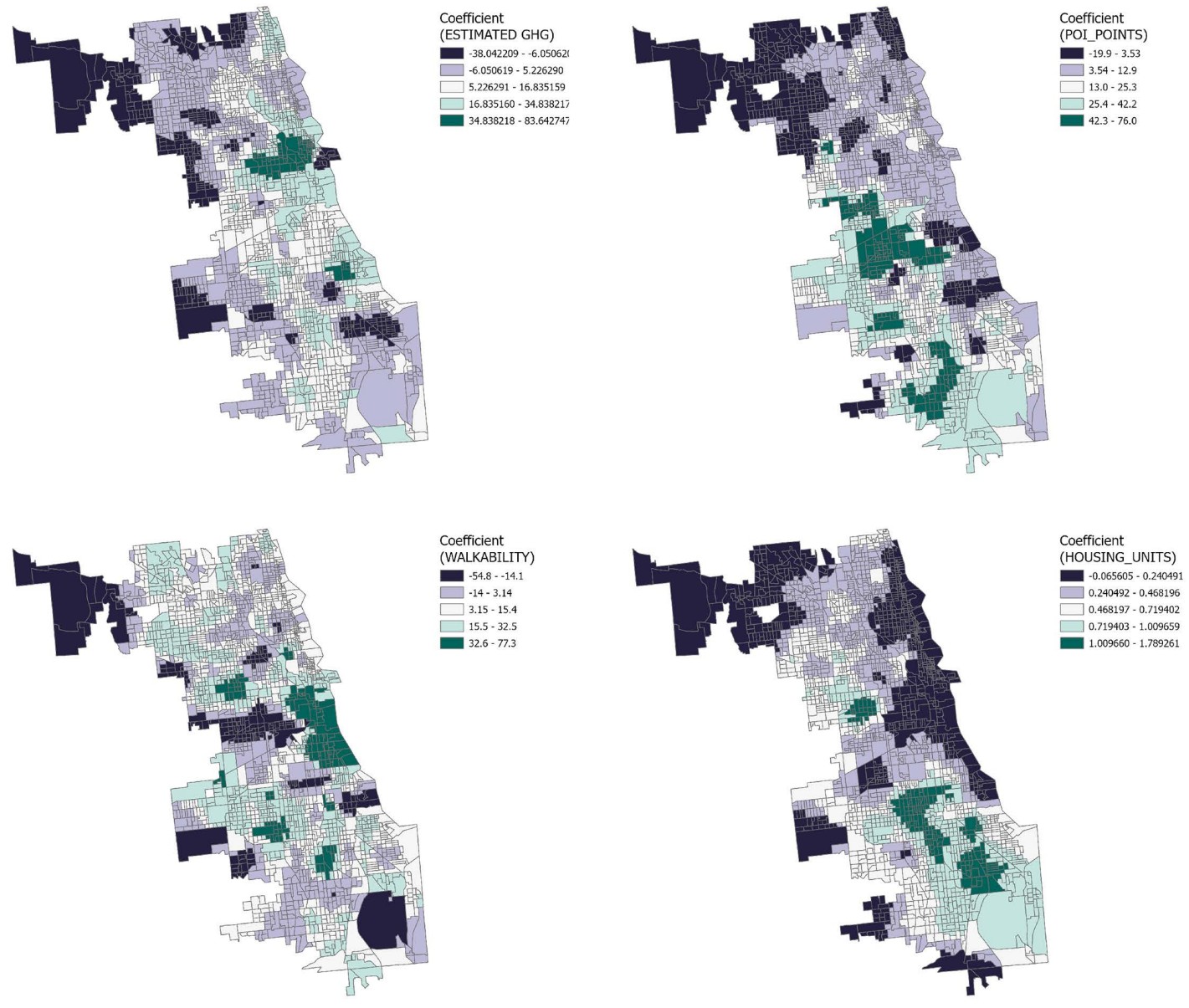

**Fig 9. GWR coefficient distribution of criminal damage crime.**

crime rates. However, a positive correlation with theft incidents may reflect that highly walkable areas afford criminals greater opportunities to evade capture and that emergency services like police vehicle access may be constrained.

**3.3.2. Northern Chicago trends.** Analysis from the northern part of Chicago indicates that the negative correlation between principal variables and crime rates may underscore the prominence of other factors. Literature suggests that these could include more stringent security measures, greater socio-economic stability, and advanced economic development. The generally affluent northern sector, with its high level of economic prosperity, may foster enhanced public safety infrastructure and community services, thereby lowering crime rates. Furthermore, the improved income levels and social services brought about by economic development could further fortify residents' sense of security and decrease the

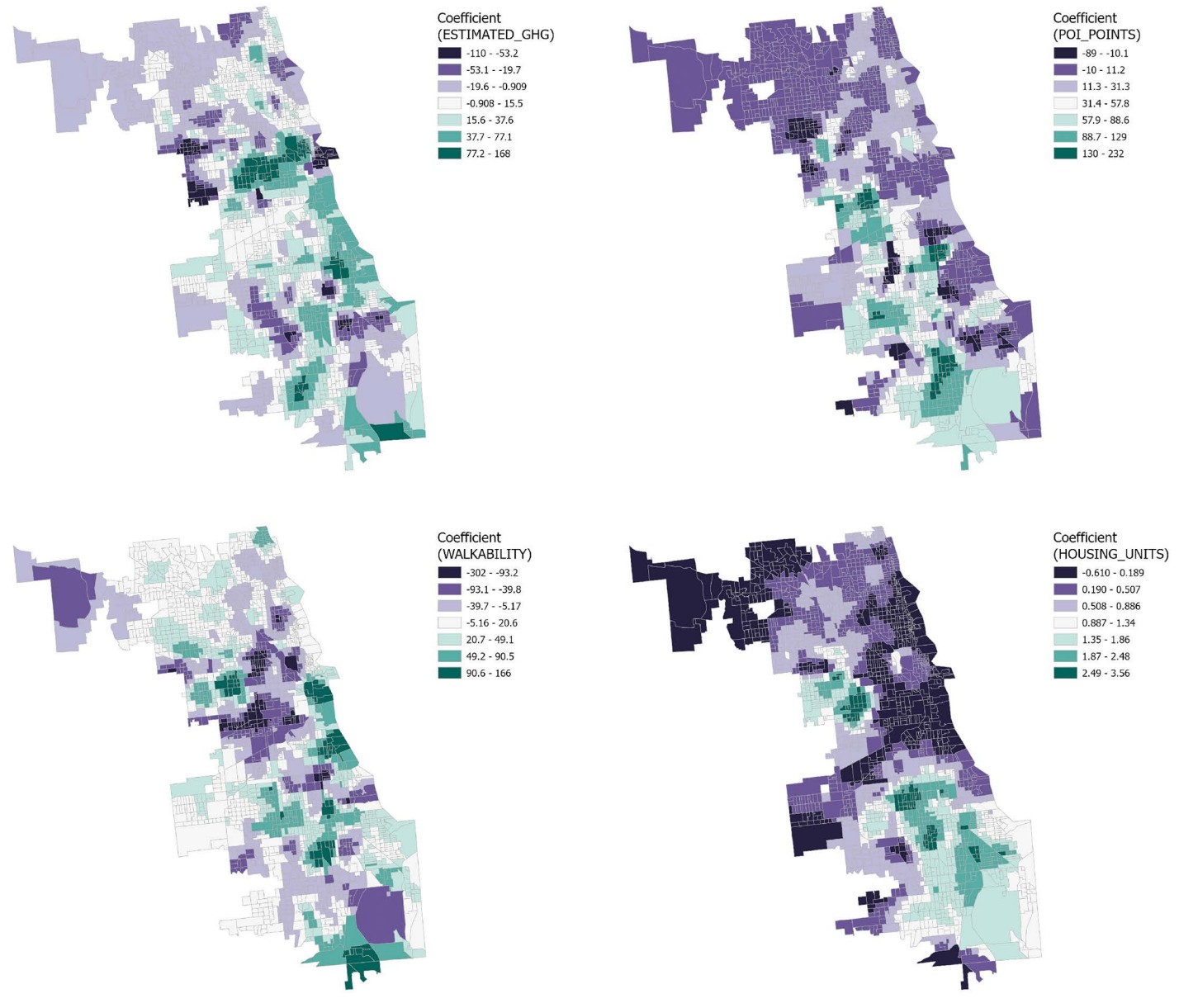

**Fig 10. GWR coefficient distribution of Battery Crime.**

incentives for crime. Thus, while there are common variables affecting crime rates, the unique socio-economic backdrop of Northern Chicago could play a determinative role in mitigating crime.

**3.3.3. Southern Chicago dynamics.** In the South Chicago area, although certain communities exhibit a correlation between the number of Housing Units, greenhouse gas emissions, and points of interest with crime incidents, overall, these variables show a weak correlation with crime, trending towards lower values. This finding suggests that the crime patterns in South Chicago may diverge from those in other areas, with factors not included in the current analysis possibly having a more significant impact on crime rates. Therefore, further exploration and analysis are warranted to unveil other potential variables influencing crime patterns in these communities.

## 4. Discussions

Our analysis shows there are significant discrepancies between media crime reporting and actual crime patterns in Chicago. Theft, criminal damage, and battery – the most frequent crimes – are systematically underrepresented in local news media, while homicides and assaults receive disproportionate coverage despite their lower frequency. This bias aligns with broader trends documented by the Pew Research Center [52], where only 11% of Americans encounter white-collar crime news compared to 37% for property crimes. Research by Velásquez et al. [53] demonstrates how media attention on sensational crimes heightens public fear, creating a perception gap where public understanding diverges from actual crime patterns.

Also, spatial bias compounds this distortion, with coverage concentrated in downtown, affluent north side, and stigmatized south side areas, reinforcing territorial stigma while neglecting other neighborhoods. Such bias likely reflects economic incentives rather than deliberate editorial choices.

Our spatial analysis using OLS and GWR models highlights how environmental and socio-economic factors' explanatory power vary in Chicago. Theft shows the strongest model fit (adjusted $R^2 = 0.552$). Dense road networks correlate positively with all crime types, as accessibility facilitates both criminal opportunities and escape routes [54,55]. However, walkability shows contrasting effects: positively associated with theft but negatively with criminal damage and battery. This paradox reflects how increased foot traffic enhances natural surveillance and community cohesion [56], while simultaneously providing opportunities for motivated offenders [57]. Similarly, building density correlates positively with all crime types, though this relationship likely reflects underlying socio-economic conditions rather than direct causation [58–63].

GWR results underscore spatial heterogeneity: downtown and northern areas show negative correlations between housing density, economic activity, and crime, likely due to strong surveillance and stability, while weaker patterns in the south suggest other unmeasured drivers.

These findings have implications for urban planning and policy development. While environmental modifications may help reduce crime risks, they should not be viewed as prescriptive solutions. Infrastructure decisions must balance crime prevention with other urban functions, such as transit efficiency and emergency response accessibility. Similarly, while vacant building remediation shows promise for crime reduction, large-scale interventions require careful consideration of budgetary constraints and community input.

The media-reality gap suggests opportunities for collaborative approaches that integrate public safety dashboards and fact-based reporting without undermining journalism's commercial imperatives. Advanced spatial analysis tools like GWR can help policymakers understand local crime variations and tailor interventions to specific neighborhood contexts, ensuring responses are appropriate to diverse socio-economic and environmental conditions.

Finally, fostering more responsible crime reporting in Chicago requires media outlets to emphasize context and accuracy over sensationalism. This involves situating incidents within broader crime trends, highlighting structural drivers, and amplifying community-led initiatives – for example, by adding a dedicated section that consistently covers these issues. By focusing on solutions and providing balanced perspectives, journalists can inform the public without fueling unnecessary fear. Engaging with diverse voices and experts can ensure nuanced portrayals of crime, encouraging constructive dialogue and evidence-based policy responses.

### 4.4. Future work

The insights gained from this study also point towards the necessity for future research to probe deeper into the causal factors behind the observed spatial patterns. For instance, sufficient research has concentrated on modifying the urban landscape to decrease crime by altering the circumstances encountered by both victims and perpetrators [64–66]. This involves investigating the roles of urban design, socioeconomic disparities, and other contributing factors in shaping the geography of crime in the Chicago area. By doing so, we can begin to design urban environments that are not only responsive to the current landscape of urban crime but also proactive in preventing its occurrence in Chicago. Such a

strategy has been theorized and proved effective [67,68]. Future research should delve deeper into the mechanics behind these spatial patterns.

## Author contributions

**Conceptualization:** Yuxuan Cai, Anzhi Chen, Ziheng Tang, Yuhan Wang, Yang Song.

**Data curation:** Yuxuan Cai, Anzhi Chen, Ziheng Tang, Yuhan Wang.

**Formal analysis:** Yuxuan Cai, Anzhi Chen.

**Investigation:** Yuxuan Cai, Anzhi Chen.

**Methodology:** Yuxuan Cai, Anzhi Chen.

**Project administration:** Yuxuan Cai, Anzhi Chen, Yang Song.

**Software:** Yuxuan Cai, Anzhi Chen.

**Supervision:** Yang Song.

**Visualization:** Anzhi Chen.

**Writing – original draft:** Yuxuan Cai, Anzhi Chen, Ziheng Tang, Yuhan Wang.

**Writing – review & editing:** Yuxuan Cai, Anzhi Chen, Yang Song.

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
