## [Decision Letter · Decision Letter 0]

1 Nov 2024

PONE-D-24-22516
Advancing Urban Management: Integrating GIS, LLMs, and Media Narratives into Environmental and Socio-economic Analyses for Enhanced Urban Crime Analysis
PLOS ONE

Dear Dr. Cai,

Thank you for submitting your manuscript to PLOS ONE. After careful consideration, we feel that it has merit but does not fully meet PLOS ONE’s publication criteria as it currently stands. Therefore, we invite you to submit a revised version of the manuscript that addresses the points raised during the review process.

We look forward to receiving your revised manuscript.

Kind regards,

Giacomo Fiumara, PhD

Academic Editor

PLOS ONE

2. We note that you have indicated that there are restrictions to data sharing for this study. For studies involving human research participant data or other sensitive data, we encourage authors to share de-identified or anonymized data. However, when data cannot be publicly shared for ethical reasons, we allow authors to make their data sets available upon request. For information on unacceptable data access restrictions, please see http://journals.plos.org/plosone/s/data-availability#loc-unacceptable-data-access-restrictions. Before we proceed with your manuscript, please address the following prompts: a) If there are ethical or legal restrictions on sharing a de-identified data set, please explain them in detail (e.g., data contain potentially identifying or sensitive patient information, data are owned by a third-party organization, etc.) and who has imposed them (e.g., a Research Ethics Committee or Institutional Review Board, etc.). Please also provide contact information for a data access committee, ethics committee, or other institutional body to which data requests may be sent. b) If there are no restrictions, please upload the minimal anonymized data set necessary to replicate your study findings to a stable, public repository and provide us with the relevant URLs, DOIs, or accession numbers. Please see http://www.bmj.com/content/340/bmj.c181.long for guidelines on how to de-identify and prepare clinical data for publication. For a list of recommended repositories, please see https://journals.plos.org/plosone/s/recommended-repositories. You also have the option of uploading the data as Supporting Information files, but we would recommend depositing data directly to a data repository if possible. Please update your Data Availability statement in the submission form accordingly.

Additional Editor Comments:

Dear authors,

The first round of reviews of the manuscript "Advancing Urban Management: Integrating GIS, LLMs, and Media Narratives into Environmental and Socio-economic Analyses for Enhanced Urban Crime Analysis" has now ended. The opinion of the Reviewers, and mine, is that the manuscript cannot be accepted in its present form and could be considered for publication in PLOS ONE if it undergoes a minor revision.

Reviewers' comments:

Reviewer's Responses to Questions

**Comments to the Author**

1. Is the manuscript technically sound, and do the data support the conclusions?

Reviewer #1: Yes

Reviewer #2: Yes

2. Has the statistical analysis been performed appropriately and rigorously?

Reviewer #1: Yes

Reviewer #2: Yes

3. Have the authors made all data underlying the findings in their manuscript fully available?

Reviewer #1: No

Reviewer #2: Yes

4. Is the manuscript presented in an intelligible fashion and written in standard English?

Reviewer #1: Yes

Reviewer #2: Yes

5. Review Comments to the Author

Reviewer #1: Comments:

1- There are some typo errors. For example, see Fig. 5. The local Moreland index. I think you mean 'Local Moran's I'.

2-Please upload high-quality images and maps in the manuscript. Numerous figures in the file that was uploaded cannot be read.

3-Where and how large a language model was used? I did not see anything in the methodology section or results. The situation is unclear.

4- I think the discussion section is weak. It is very useful to mention your findings, compare them with previous study findings, and discuss them. For example, I think several studies have been conducted about crimes in Chicago in recent years.

Best

Reviewer #2: The paper presents an original and comprehensive study that integrates Geographic Information Systems (GIS), Large Language Models (LLMs), and media narratives to examine urban crime dynamics in Chicago. This interdisciplinary approach, combining advanced spatial and machine learning tools, makes significant contributions to urban studies, media analysis, and crime prevention strategies. Overall, the study is well-constructed, but minor revisions are necessary to enhance clarity and focus in some sections before it can be considered for publication.

Originality

The integration of GIS, LLMs, and media narratives is a novel approach in the field of urban crime analysis. The study stands out by addressing the interplay between environmental, socioeconomic factors, and media reporting, which provides a fresh perspective compared to traditional crime studies. The originality is clear, and I was unable to find significant overlap in other published papers.

Abstract

The abstract effectively summarizes the research objectives, methodology, and findings. However, the term "GIS" is used without prior explanation. I recommend that the authors introduce "Geographic Information System" before using the acronym, ensuring clarity for readers unfamiliar with the term. Other than this, the abstract provides a concise overview of the study’s key points.

Introduction

The introduction offers a detailed and well-researched background on the relationship between media reporting and crime perception. It successfully sets up the study’s objectives, but Section 1.1 is somewhat verbose and could be condensed into a single paragraph. Streamlining this part will improve readability while retaining the critical points. Additionally, when discussing correlations between factors such as temperature and crime, the distinction between correlation and causation should be more clearly stated to avoid any potential misinterpretation.

Methodology

The methodology is one of the manuscript's strongest sections, providing a clear and structured approach. The use of GIS crime data, combined with media reporting analysis using GPT-3.5, is well-executed. The authors’ innovative application of LLMs to classify crime reports with high accuracy is commendable, and the spatial analysis using Geographically Weighted Regression (GWR) is solidly grounded in spatial econometrics. Overall, this section demonstrates rigorous technical standards.

Results

The results section is comprehensive and provides clear insights into the discrepancies between media-reported and actual crime data. The authors effectively highlight key findings, such as the media’s overrepresentation of certain crimes (e.g., homicide) and underreporting of others (e.g., battery). However, distinguishing between different levels of theft (e.g., petty vs. grand theft) would offer more nuance to the analysis of surveillance impact.

Discussion

The discussion is well-structured, synthesizing the results with broader literature. The spatial analysis findings are plausible and align well with the study’s hypotheses. However, the policy recommendations, while relevant, could be more specific. The suggestions currently offered are somewhat general and would benefit from more targeted advice based on the study’s spatial findings. Providing more actionable insights would enhance the practical application of the study’s conclusions for urban planners and policymakers.

Data and Code Availability

The transparency in making the environmental, socioeconomic, and media data available through publicly accessible sources like the Chicago Data Portal and Perigon API is commendable. The authors’ adherence to open science practices ensures that the study can be replicated, which is a strength of the manuscript.

Recommendation

The manuscript presents a novel and well-executed study that integrates multiple analytical tools to examine urban crime dynamics. The study is methodologically sound and contributes valuable insights to both academic and practical discussions on urban management. However, I recommend minor revisions to improve clarity and expand on certain sections, particularly in explaining correlations, streamlining the introduction, and enhancing the specificity of policy recommendations. Once these revisions are made, the paper will be well-suited for publication in PLOS ONE.

6. PLOS authors have the option to publish the peer review history of their article (what does this mean?). If published, this will include your full peer review and any attached files.

Reviewer #1: **Yes: **Alireza Mohammadi

Reviewer #2: No

---

## [Author Response · Author response to Decision Letter 1]

6 Nov 2024

Reviewer #1:

- Comments: There are some typo errors. For example, see Fig. 5. The local Moreland index. I think you mean 'Local Moran's I'.

- Thank you for pointing this out. We have corrected the typo errors, including the reference to the 'Local Moran's I' in Figure 5. Additionally, we carefully reviewed the manuscript to address any remaining language issues, ensuring improved clarity and readability.

- 2-Please upload high-quality images and maps in the manuscript. Numerous figures in the file that was uploaded cannot be read.

- Thank you for your feedback. We have re-uploaded the high-resolution versions of all images to ensure clarity and readability.

- 3-Where and how large a language model was used? I did not see anything in the methodology section or results. The situation is unclear.

- We have implemented few-shot learning methods for text classification of media-reported crimes and provided a detailed explanation in the newly renamed “Data and Methods” section. Specifically, subsection 2.2 is now titled “Use of LLM in Media Reporting” and includes comprehensive details on applying LLMs in our study. The classification results are utilized to compare media-reported data with actual crime data in section 3.1, “Comparison Between Media Data and Crime Data,” enhancing the clarity and depth of our analysis.

- 4- I think the discussion section is weak. It is very useful to mention your findings, compare them with previous study findings, and discuss them. For example, I think several studies have been conducted about crimes in Chicago in recent years.

- Thank you for your valuable suggestion. We have incorporated references to recent studies on the impact of Chicago's environment on crime, as well as research on how media-reported crimes influence public perception. These references are now included in the discussion section to connect our findings with existing literature, thereby providing a more comprehensive context for our results.

Reviewer #2:

Originality

- The integration of GIS, LLMs, and media narratives is a novel approach in the field of urban crime analysis. The study stands out by addressing the interplay between environmental, socioeconomic factors, and media reporting, which provides a fresh perspective compared to traditional crime studies. The originality is clear, and I was unable to find significant overlap in other published papers.

- Thank you for your feedback! We are glad that you appreciate our work.

Abstract

- The abstract effectively summarizes the research objectives, methodology, and findings. However, the term "GIS" is used without prior explanation. I recommend that the authors introduce "Geographic Information System" before using the acronym, ensuring clarity for readers unfamiliar with the term. Other than this, the abstract provides a concise overview of the study’s key points.

- Thank you for your feedback. We have addressed this issue.

Introduction

- The introduction offers a detailed and well-researched background on the relationship between media reporting and crime perception. It successfully sets up the study’s objectives, but Section 1.1 is somewhat verbose and could be condensed into a single paragraph. Streamlining this part will improve readability while retaining the critical points. Additionally, when discussing correlations between factors such as temperature and crime, the distinction between correlation and causation should be more clearly stated to avoid any potential misinterpretation.

- Thank you for pointing this out. We have substantially rewritten Section 1.1 to improve readability by making it more concise while keeping the original information as much as possible. We also add contents to distinguish the correlation and causation in the Introduction part.

Methodology

- The methodology is one of the manuscript's strongest sections, providing a clear and structured approach. The use of GIS crime data, combined with media reporting analysis using GPT-3.5, is well-executed. The authors’ innovative application of LLMs to classify crime reports with high accuracy is commendable, and the spatial analysis using Geographically Weighted Regression (GWR) is solidly grounded in spatial econometrics. Overall, this section demonstrates rigorous technical standards.

- Thank you for your appreciation.

Results

- The results section is comprehensive and provides clear insights into the discrepancies between media-reported and actual crime data. The authors effectively highlight key findings, such as the media’s overrepresentation of certain crimes (e.g., homicide) and underreporting of others (e.g., battery). However, distinguishing between different levels of theft (e.g., petty vs. grand theft) would offer more nuance to the analysis of surveillance impact.

- Thank you for pointing out the issue. We have corrected the section title from "Results" to "Discussion."

Discussion

- The discussion is well-structured, synthesizing the results with broader literature. The spatial analysis findings are plausible and align well with the study’s hypotheses. However, the policy recommendations, while relevant, could be more specific. The suggestions currently offered are somewhat general and would benefit from more targeted advice based on the study’s spatial findings. Providing more actionable insights would enhance the practical application of the study’s conclusions for urban planners and policymakers..

- Thank you for pointing this out. We agree that the policy recommendations could be more specific and actionable. We had revised this section to include more detailed, targeted advice.

Data and Code Availability

The transparency in making the environmental, socioeconomic, and media data available through publicly accessible sources like the Chicago Data Portal and Perigon API is commendable. The authors’ adherence to open science practices ensures that the study can be replicated, which is a strength of the manuscript.

- Thank you for your appreciation.

Recommendation

The manuscript presents a novel and well-executed study that integrates multiple analytical tools to examine urban crime dynamics. The study is methodologically sound and contributes valuable insights to both academic and practical discussions on urban management. However, I recommend minor revisions to improve clarity and expand on certain sections, particularly in explaining correlations, streamlining the introduction, and enhancing the specificity of policy recommendations. Once these revisions are made, the paper will be well-suited for publication in PLOS ONE.

- Thank you for your constructive feedback. We have revised the manuscript to enhance the explanation of the correlations and streamlined the introduction for better clarity. Additionally, we have expanded the specificity of our policy recommendations, particularly for different community groups in the Chicago area, including suggestions from media and community stakeholders.

---

## [Decision Letter · Decision Letter 1]

13 May 2025

PONE-D-24-22516R1
Advancing Urban Management: Integrating GIS, LLMs, and Media Narratives into Environmental and Socio-economic Analyses for Enhanced Urban Crime Analysis
PLOS ONE

Dear Dr. Cai,

Thank you for submitting your manuscript to PLOS ONE. After careful consideration, we feel that it has merit but does not fully meet PLOS ONE’s publication criteria as it currently stands. Therefore, we invite you to submit a revised version of the manuscript that addresses the points raised during the review process.
 
Please make sure that you carefully and thoroughly address all comments by reviewer 3, and that you provide a detailed response to them.

We look forward to receiving your revised manuscript.

Kind regards,

Federico Botta

Academic Editor

PLOS ONE

Reviewers' comments:

Reviewer's Responses to Questions

**Comments to the Author**

1. If the authors have adequately addressed your comments raised in a previous round of review and you feel that this manuscript is now acceptable for publication, you may indicate that here to bypass the “Comments to the Author” section, enter your conflict of interest statement in the “Confidential to Editor” section, and submit your "Accept" recommendation.

Reviewer #1: All comments have been addressed

Reviewer #2: All comments have been addressed

Reviewer #3: All comments have been addressed

2. Is the manuscript technically sound, and do the data support the conclusions?

Reviewer #1: Yes

Reviewer #2: Yes

Reviewer #3: No

3. Has the statistical analysis been performed appropriately and rigorously?

Reviewer #1: Yes

Reviewer #2: Yes

Reviewer #3: No

4. Have the authors made all data underlying the findings in their manuscript fully available?

Reviewer #1: Yes

Reviewer #2: Yes

Reviewer #3: Yes

5. Is the manuscript presented in an intelligible fashion and written in standard English?

Reviewer #1: Yes

Reviewer #2: Yes

Reviewer #3: Yes

6. Review Comments to the Author

Reviewer #1: I am pleased to confirm that the authors have successfully addressed all the comments and suggestions I previously provided.

Reviewer #2: The authors have addressed all the comments I raised in the previous review. The introduction of GIS is now clearer, the results include more commentary about different nuances of crimes, and the policy recommendations are more specific and practical. The manuscript is technically sound, and the methods and data are well-explained. I think the paper is now well-written and ready to be published.

Reviewer #3: Advancing Urban Management

The manuscript raises interesting points and explores connections between crime reporting, urban design, and socio-economic variables. I am aware that there has been another round of reviewers before me and their comments have been addressed. However, I have serious concerns regarding the presentation and rationale for methodological choices. I am concerned about how the data is analysed and the interpretation of the results. The policy recommendations must be seriously considered and analyse if this is a result of the research. Below I give the authors some feedback and suggest a redesign of their study and submit the manuscript elsewhere.

First, regarding the variables.

Regarding the points of Interest, the high heterogeneity in POI counts suggests a potential data issue. The locations with 400+ POIs seems wrong. Or the data has some POI that are less relevant than others.

Population counts of 10,000+ people in a small area? This raise questions about the data's validity. How is there significant population in that location, but without corresponding housing?

For the water area, elaborate on its role in the analysis and how it contributes to the study's aesthetic dimension. Some variables, like the southern lake in Chicago shown on the map, appear disproportionately large. Clarify what these variables represent. And what is the rationale. It cannot be that I just push the data into some ML algorithm, and then something comes out so it must be true. Why water? How would this look like in a non-coastal city?

The rationale of the Greenhouse Gas Emissions as Proxy for activity levels seems incorrect (like a fancy hotel or restaurant has little GHG, but a factory has loads). Clarify whether this is your contribution or based on prior research. If the latter, cite the appropriate source. If original, provide justification for its use in this context.

Second, regarding some methods.

Some terms are introduced without any intuition, reference or context, for example, the Cohen's Kappa. I suggest an intuitive explanation of Cohen's kappa, its relevance to the study, and why a value of 0.92 is considered high. Additionally, include a citation demonstrating its application in this context.

The same goes with the Perigon Database. A brief explanation of what the Perigon database is, how it is compiled, and its relevance to crime research. Are all news events included in this database? Add a citation to support its use in this field.

Categorization of Crimes. Why is it necessary to categorize reported crimes into designated categories? What are the "three distinct crime typologies," and who determined these categories? Clearly define and justify the choice of categories with proper references.

For concepts such as "surveillance" (measured by road network density), clarify whether this metric is based on prior work (with a proper citation) or is a novel contribution. If it is the second, explicitly state that the metric is proposed as a proxy to study surveillance. The same applies to "maintenance" (using the National Walkability Index) and "total water area" (linked to aesthetics).

Regarding the Socio-Economic Characteristics, there is a phrase like "Socio-economic characteristics have been extensively discussed in previous re-search, demonstrating a significant correlation with crime rates." lacks a citation. Well, can you actually say that? (And research is a single word).

Third, regarding the results and conclusions

The policy recommendations are basically good wishes but far from the scope of the manuscript and the results, and with a lack of academic rigour. The findings do not really suggest how to prevent crime, but they present some correlation with media. This is fine. One cannot solve everything in a single paper. However, making policy recommendations is not just adding one last paragraph to your manuscript. Phrases like "limit excessive connectivity" make me feel like the implications were not considered seriously and were added just to say that it has policy implications. Yes, cities could "restrict potential escape routes for offenders", but what budget is needed? What implications in terms of road traffic would this proposal have? Would a bus route be blocked by this idea? Can ambulances or firefighters travel through the city in case of an emergency?

The same with "media outlets should...". Media is a business, and as such, its objective is to make money. If only 1/1000 crimes appear in the news, should they start reporting minor thefts over a mass shooting, only to be more accurate?

Minor comments

All equations are punctuated and integrated into the text following proper academic conventions. See https://www2.math.uconn.edu/~khlee/math2784s10/writingtips.pdf

The map with a black background is inconsistent with others and difficult to interpret. Revise for consistency and readability.

The results follow a weird narrative. Why start with the number of crimes across months, if there is almost no variation across time? Present results from general to specific for better narrative flow. Start with the total number of crimes reported in the media, then compare media reports to actual crime data, followed by geographical mapping. The time series analysis, while interesting, conveys minimal new insights and might be better placed in supplementary materials.

7. PLOS authors have the option to publish the peer review history of their article (what does this mean?). If published, this will include your full peer review and any attached files.

Reviewer #1: **Yes: **Alireza Mohammadi, Department of Geography and Urban Planning, Faculty of Social Sciences, University of Mohaghegh Ardabili, Azerbaijan, Ardabil, Iran

Reviewer #2: No

Reviewer #3: No

---

## [Author Response · Author response to Decision Letter 2]

19 Jul 2025

Response to Reviewer 1 Comments

Point 1.1:

Reviewer #1: I am pleased to confirm that the authors have successfully addressed all the comments and suggestions I previously provided.

Response 1.1: Thank you for your recognition.

Response to Reviewer 2 Comments

Point 2.1:

Reviewer #2: The authors have addressed all the comments I raised in the previous review. The introduction of GIS is now clearer, the results include more commentary about different nuances of crimes, and the policy recommendations are more specific and practical. The manuscript is technically sound, and the methods and data are well-explained. I think the paper is now well-written and ready to be published.

Response 2.1: Thank you for your recognition and for your suggestion in improving this paper.

Response to Reviewer 3 Comments

Reviewer #3: Advancing Urban Management

The manuscript raises interesting points and explores connections between crime reporting, urban design, and socio-economic variables. I am aware that there has been another round of reviewers before me and their comments have been addressed. However, I have serious concerns regarding the presentation and rationale for methodological choices. I am concerned about how the data is analysed and the interpretation of the results. The policy recommendations must be seriously considered and analyse if this is a result of the research. Below I give the authors some feedback and suggest a redesign of their study and submit the manuscript elsewhere.

Point 3.1:

First, regarding the variables.

Regarding the points of Interest, the high heterogeneity in POI counts suggests a potential data issue. The locations with 400+ POIs seems wrong. Or the data has some POI that are less relevant than others.

Response 3.1:

Thank you for raising this concern. The POI data in our study were sourced from OpenStreetMap, which includes a wide range of features such as restaurants, retail outlets, public amenities, and service points. Locations with over 400 POIs primarily correspond to downtown Chicago and other commercial hubs where such density is realistic given the concentration of businesses and services. These high POI counts reflect the actual urban heterogeneity rather than data errors. We retained all POIs because our objective was to capture the full scope of territoriality and activity support, as emphasized in CPTED theory. To avoid misinterpretation, we will add a clarification in the Methods section 2.3.1 to indicate that while some POIs may be less directly relevant to crime dynamics, their inclusion ensures consistency and comprehensiveness in measuring environmental complexity.

Point 3.2: Population counts of 10,000+ people in a small area? This raise questions about the data's validity. How is there significant population in that location, but without corresponding housing?

Response 3.2:

Thank you for mentioning this point. We re-examined the issues with the code and the original data. Previously, the display problem was due to problems with the legend range display of the regional values in the code. We re-output all the visualization results based on the gis software.

Point 3.3: For the water area, elaborate on its role in the analysis and how it contributes to the study's aesthetic dimension. Some variables, like the southern lake in Chicago shown on the map, appear disproportionately large. Clarify what these variables represent. And what is the rationale. It cannot be that I just push the data into some ML algorithm, and then something comes out so it must be true. Why water? How would this look like in a non-coastal city?

Response 3.3: Thank you for your suggestions and they are indeed valuable. The inclusion of water area is grounded in Crime Prevention through Environmental Design, specifically the Image Maintenance dimension, which relates to environmental aesthetics and quality. Water bodies in urban environments, such as Lake Michigan and riverfront areas in Chicago mentioned in our article, typically are associated with improved walkability, increased recreational activity, and natural surveillance, all of which can influence crime dynamics. Even if these areas may appear disproportionately large on maps, they represent the actual geographic features that shape urban design and usage patterns since Chicago is next to the lake. In non-coastal cities, comparable variables would include reservoirs, riverfronts, or major parks, as the underlying principle is to capture open space aesthetics and their effect on human activity. This is not a purely data-driven inclusion; rather, it reflects how environmental quality impacts crime risk, especially the quality of coastal infrastructure. We will clarify this rationale in the Methods section, at the end of 2.3.1, for generalizability and clarity.

Point 3.4:

The rationale of the Greenhouse Gas Emissions as Proxy for activity levels seems incorrect (like a fancy hotel or restaurant has little GHG, but a factory has loads). Clarify whether this is your contribution or based on prior research. If the latter, cite the appropriate source. If original, provide justification for its use in this context.

Response 3.4: Thank you for this comment. We have clarified the rationale for using GHG emissions as a proxy for urban economic and activity intensity, referencing prior studies that link emissions with land-use and mobility patterns. We also acknowledge its limitations (e.g., industrial zones vs. commercial areas) and added this discussion, along with a note suggesting future integration of direct mobility measures, to the Methods and Limitations sections.

Point 3.5: Second, regarding some methods.

Some terms are introduced without any intuition, reference or context, for example, the Cohen's Kappa. I suggest an intuitive explanation of Cohen's kappa, its relevance to the study, and why a value of 0.92 is considered high. Additionally, include a citation demonstrating its application in this context.

Response 3.5: Thank you and this is a helpful suggestion. We’ve added an intuitive explanation of Cohen’s kappa in the Methods section and provided citations to describe its interpretation. We also clarify that our κ = 0.92 falls within the “almost perfect” agreement range.

Point 3.6: The same goes with the Perigon Database. A brief explanation of what the Perigon database is, how it is compiled, and its relevance to crime research. Are all news events included in this database? Add a citation to support its use in this field.

Response 3.6: Thank you for the suggestion. We clarified the details about Perigon database.

Point 3.7: Categorization of Crimes. Why is it necessary to categorize reported crimes into designated categories? What are the "three distinct crime typologies," and who determined these categories? Clearly define and justify the choice of categories with proper references.

Response 3.7: Thank you for raising this point. The categorization of reported crimes follows the official classification used by the Chicago Data Portal, which aligns with the Illinois Uniform Crime Reporting (UCR) system. This ensures consistency between media reports and official crime data, enabling comparative analysis. The “three distinct crime typologies” (theft, criminal damage, and battery) were selected based on their prevalence in the Chicago Open Data dataset. We have clarified this in the Methods section 2.2 and added citations to the official data source.

Point 3.8: For concepts such as "surveillance" (measured by road network density), clarify whether this metric is based on prior work (with a proper citation) or is a novel contribution. If it is the second, explicitly state that the metric is proposed as a proxy to study surveillance. The same applies to "maintenance" (using the National Walkability Index) and "total water area" (linked to aesthetics).

Response 3.8: Thank you for raising these questions. These variables are mainly based on summarizations from previous studies, and we have added citations to corresponding research. Research shows that street connectivity is an important physical environmental feature for understanding criminal opportunities and guardianship, and enhancing road network connectivity and density can increase pedestrian and vehicle activity, boost natural surveillance and reduce theft opportunities. Natural surveillance maximizes the visibility of spaces and their users through the design layout of physical features, activities, and people.

Walkability as an environmental maintenance indicator is based on CPTED's Image/Maintenance principle. The "broken windows theory" emphasizes that "neglected areas invite crime, which reinforces the need for good property maintenance to assert visible ownership of space." Maintenance focuses on keeping areas clean and well-maintained. Places with litter, graffiti, and overgrown landscaping send a signal that no one cares what happens in that area. The use of water area as an environmental quality indicator falls under exploratory research. Although existing studies mainly focus on the health benefits and socio-economic impacts of urban blue spaces, there is a lack of literature directly studying the relationship between water bodies and land crimes. Our analysis is based on theoretical assumptions regarding environmental aesthetics and the potential impact of open spaces on community cohesion.

Point 3.9: Regarding the Socio-Economic Characteristics, there is a phrase like "Socio-economic characteristics have been extensively discussed in previous re-search, demonstrating a significant correlation with crime rates." lacks a citation. Well, can you actually say that? (And research is a single word).

Response 3.9: Thank you for providing the feedback. We corrected the typo (“research”) and added citations to substantiate the claim that socio-economic characteristics are strongly associated with crime rates in prior studies.

Point 3.10 Third, regarding the results and conclusions

The policy recommendations are basically good wishes but far from the scope of the manuscript and the results, and with a lack of academic rigour. The findings do not really suggest how to prevent crime, but they present some correlation with media. This is fine. One cannot solve everything in a single paper. However, making policy recommendations is not just adding one last paragraph to your manuscript. Phrases like "limit excessive connectivity" make me feel like the implications were not considered seriously and were added just to say that it has policy implications. Yes, cities could "restrict potential escape routes for offenders", but what budget is needed? What implications in terms of road traffic would this proposal have? Would a bus route be blocked by this idea? Can ambulances or firefighters travel through the city in case of an emergency?

Response 3.10: Thank you for this point. We agree that our original policy recommendations lacked sufficient grounding in the study’s findings. We have revised the first two paragraphs in Section 4.3 to ensure recommendations are explicitly tied to the evidence presented in the Results and Discussion sections. Instead of general prescriptions, we now provide nuanced interpretations of how observed spatial and socio-economic correlations might inform planning strategies, while removed prescriptive statements.

Point 3.11. The same with "media outlets should...". Media is a business, and as such, its objective is to make money. If only 1/1000 crimes appear in the news, should they start reporting minor thefts over a mass shooting, only to be more accurate?

Response 3.11: Thank you for this point. We acknowledge that the suggestions we provided were immature and that the media operates under market-driven incentives, which makes it unrealistic to expect equal coverage of all crime types. Our intention was not to prescribe media practices, but to highlight the societal implications of selective reporting. We have revised this section to remove normative language and instead focus on the theoretical relevance of media narratives shaping public perception and policy. We now frame this as an observation of structural constraints rather than a recommendation for content redistribution.

Point 3.12. Minor comments

All equations are punctuated and integrated into the text following proper academic conventions. See https://www2.math.uconn.edu/~khlee/math2784s10/writingtips.pdf

Response 3.12:

Thank you for your suggestion. We have adjusted the table and description of the formula based on the standards of previous studies.

Point 3.13. The map with a black background is inconsistent with others and difficult to interpret. Revise for consistency and readability.

Response 3.13:

Thank you for raising this point. We have modified and unified the background colors of all images and ensured readability.

Point 3.14. The results follow a weird narrative. Why start with the number of crimes across months, if there is almost no variation across time? Present results from general to specific for better narrative flow. Start with the total number of crimes reported in the media, then compare media reports to actual crime data, followed by geographical mapping. The time series analysis, while interesting, conveys minimal new insights and might be better placed in supplementary materials.

Response 3.14:

Thank you for mentioning this. We have revised the narrative logic at the beginning of section 3.1. Now we start the description from the general conclusion and weaken the results of the time series.

---

## [Decision Letter · Decision Letter 2]

30 Jul 2025

PONE-D-24-22516R2
Advancing Urban Management: Integrating GIS, LLMs, and Media Narratives into Environmental and Socio-economic Analyses for Enhanced Urban Crime Analysis
PLOS ONE

Dear Dr. Cai,

Thank you for submitting your manuscript to PLOS ONE. After careful consideration, we feel that it has merit but does not fully meet PLOS ONE’s publication criteria as it currently stands. Therefore, we invite you to submit a revised version of the manuscript that addresses the points raised during the review process.

We look forward to receiving your revised manuscript.

Kind regards,

Federico Botta

Academic Editor

PLOS ONE

Journal Requirements:

Reviewers' comments:

Reviewer's Responses to Questions

**Comments to the Author**

1. If the authors have adequately addressed your comments raised in a previous round of review and you feel that this manuscript is now acceptable for publication, you may indicate that here to bypass the “Comments to the Author” section, enter your conflict of interest statement in the “Confidential to Editor” section, and submit your "Accept" recommendation.

Reviewer #3: All comments have been addressed

2. Is the manuscript technically sound, and do the data support the conclusions?

Reviewer #3: Yes

3. Has the statistical analysis been performed appropriately and rigorously?

Reviewer #3: Yes

4. Have the authors made all data underlying the findings in their manuscript fully available?

Reviewer #3: Yes

5. Is the manuscript presented in an intelligible fashion and written in standard English?

Reviewer #3: Yes

6. Review Comments to the Author

Reviewer #3: The authors modified the manuscript following all my previous suggestions. I only have a few comments for them.

Please check the following citations since they have a typo:

[22] resi-dential

[25] Lan-guage

[29] kohen (is a name, so write "Kohen". Also "sage"

[54] chicago

[60] chicago, covid

[61] shiraz, iran

[62] belgium

After the expression (3), the definition of w_ij does not need the curly brackets.

The (2) for the second expression should be in the same line as the expression.

The section 4.3 is called "Policy Recommendations" but I suggest changing this subtitle. I am happy that you dropped some of the recommendations that you wrote before, but now (better) that there are no punctual recommendations, then the subtitle seems weird. And I would suggest that the whole "Discussions" section could fit on a single page, from the three pages that you have now. Usually, on a manuscript, less text but more impactful is better.

7. PLOS authors have the option to publish the peer review history of their article (what does this mean?). If published, this will include your full peer review and any attached files.

Reviewer #3: **Yes: **Rafael Prieto-Curiel

---

## [Author Response · Author response to Decision Letter 3]

17 Aug 2025

Response to Reviewer 3 Comments

Reviewer #3:

The authors modified the manuscript following all my previous suggestions. I only have a few comments for them.

Point 1.1：Please check the following citations since they have a typo:

[22] resi-dential

[25] Lan-guage

[29] kohen (is a name, so write "Kohen". Also "sage"

[54] chicago

[60] chicago, covid

[61] shiraz, iran

[62] Belgium

Response 1.1:

Thank you for pointing out these issues. We recheck and rewrite these typo issues.

Point 1.2：After the expression (3), the definition of w_ij does not need the curly brackets.

The (2) for the second expression should be in the same line as the expression.

Response 1.2: Thank you for mentioning this. We removed the curly brackets, and also fixed the order (2) place issues.

Point 1.3：The section 4.3 is called "Policy Recommendations" but I suggest changing this subtitle. I am happy that you dropped some of the recommendations that you wrote before, but now (better) that there are no punctual recommendations, then the subtitle seems weird. And I would suggest that the whole "Discussions" section could fit on a single page, from the three pages that you have now. Usually, on a manuscript, less text but more impactful is better.

Response 1.3: Thank you for your suggestion. We had rewritten the whole discussion section. And make these contexts more concise and clearer and keep the impactful thinking for future readers.

---

## [Editor Report · Decision Letter 3]

21 Aug 2025

Advancing Urban Management: Integrating GIS, LLMs, and Media Narratives into Environmental and Socio-economic Analyses for Enhanced Urban Crime Analysis

PONE-D-24-22516R3

Dear Dr. Cai,

We’re pleased to inform you that your manuscript has been judged scientifically suitable for publication and will be formally accepted for publication once it meets all outstanding technical requirements.

Kind regards,

Federico Botta

Academic Editor

PLOS ONE
---

## [Editor Report · Acceptance letter]

PONE-D-24-22516R3

PLOS ONE

Dear Dr. Cai,

I'm pleased to inform you that your manuscript has been deemed suitable for publication in PLOS ONE. Congratulations! Your manuscript is now being handed over to our production team.

Kind regards,

on behalf of

Dr. Federico Botta

Academic Editor

PLOS ONE